# Prime Editing and DNA Repair System: Balancing Efficiency with Safety

**DOI:** 10.3390/cells13100858

**Published:** 2024-05-17

**Authors:** Karim Daliri, Jürgen Hescheler, Kurt Paul Pfannkuche

**Affiliations:** 1Institute for Neurophysiology, Centre for Physiology and Pathophysiology, Medical Faculty and University Hospital of Cologne, University of Cologne, 50931 Cologne, Germanykurt.pfannkuche@uni-koeln.de (K.P.P.); 2Marga and Walter Boll-Laboratory for Cardiac Tissue Engineering, University of Cologne, 50931 Cologne, Germany; 3Center for Molecular Medicine Cologne (CMMC), University of Cologne, 50931 Cologne, Germany

**Keywords:** prime editing, PE4, PE5, MLH1, cancer

## Abstract

Prime editing (PE), a recent progression in CRISPR-based technologies, holds promise for precise genome editing without the risks associated with double-strand breaks. It can introduce a wide range of changes, including single-nucleotide variants, insertions, and small deletions. Despite these advancements, there is a need for further optimization to overcome certain limitations to increase efficiency. One such approach to enhance PE efficiency involves the inhibition of the DNA mismatch repair (MMR) system, specifically MLH1. The rationale behind this approach lies in the MMR system’s role in correcting mismatched nucleotides during DNA replication. Inhibiting this repair pathway creates a window of opportunity for the PE machinery to incorporate the desired edits before permanent DNA repair actions. However, as the MMR system plays a crucial role in various cellular processes, it is important to consider the potential risks associated with manipulating this system. The new versions of PE with enhanced efficiency while blocking MLH1 are called PE4 and PE5. Here, we explore the potential risks associated with manipulating the MMR system. We pay special attention to the possible implications for human health, particularly the development of cancer.

## 1. Introduction

The RNA-guided CRISPR (Clustered Regularly Interspaced Short Palindromic Repeats) endonuclease system was first identified in *Escherichia coli (E. coli)* in 1987, with its unique genomic structure providing evidence for its function. This structure evolved as an adaptive immune system employed by bacteria and archaea. It utilizes a set of CRISPR-associated (Cas) genes to incorporate exogenous material into the CRISPR locus. This incorporated material is then transcribed into RNA templates that guide the targeted destruction of mobile elements at the DNA or RNA level [1]. To date, three types of CRISPR systems have been identified, each with distinct mechanisms of action. Unlike the Type II CRISPR/Cas system, which uses a single endonuclease called Cas9 to identify and cleave target DNA, Type I and III systems employ a group of Cas genes to perform RNA processing, target recognition, and cleavage [1,2,3]. In the Type II system, a pair of non-coding RNAs guide the Cas9 endonuclease to its target DNA sequence. These RNAs include a CRISPR RNA (crRNA) and an auxiliary transactivating crRNA (tracrRNA). The crRNA contains two key components which are a 20-nucleotide guide sequence (often called spacer), which determines target specificity via Watson-Crick base-pairing with the target DNA and an invariant “direct repeat” component, which pairs with the “antirepeat” segment of the tracrRNA to form an RNA duplex. The paired repeat segments of both RNAs attract Cas9 to the complex [4,5].

CRISPR-Cas-based genome editing relies on four primary double-strand break (DSB) repair pathways in eukaryotic cells: non-homologous end joining (NHEJ), homologous recombination (HR), microhomology-mediated end joining (MMEJ), and single-strand annealing (SSA). Each pathway leads to different editing outcomes [6]. Therefore, researchers are actively developing strategies to control the pathway that cells use to repair DSBs during editing. However, the mechanisms governing DSB repair are intricately complex and remain incompletely understood [6,7]. A deeper understanding of these mechanisms promises fundamental insights into genome integrity and will pave the way for more sophisticated genome editing strategies.

Prime editors, as the latest generation of CRISPR-Cas9 -based technologies, can insert short DNA sequences without generating DSBs or requiring an external template. They consist of a nicking version of Cas9 fused to a reverse transcriptase (RT) domain, which is complexed with a prime editing guide RNA (pegRNA). The pegRNA consists of a primer-binding site homologous to the sequence in the target and a reverse transcriptase (RT) template, including the intended edit, all in the 3′ extension of a standard CRISPR-Cas9 guide RNA. At the target site, Cas9 nicks one strand of the DNA, which then anneals to the primer-binding site on the pegRNA, extending by the Cas9-fused reverse transcriptase (RT) using the pegRNA-encoded template sequence. Next, DNA repair mechanisms (miss match repair, MMR) resolve the conflicting sequences on the two DNA strands, ultimately writing the intended edit into the genome [8]. The prime editing (PE) process is complex, and the factors influencing its efficacy are not fully understood. It involves multiple somewhat independent stages, including three DNA binding events and successful MMR DNA repair, all necessary for successful editing. Each stage is potentially influenced by the edited sequence. Furthermore, the presence of a nick on the edited strand attracts the MMR machinery to that DNA strand, initiating the EXO1-mediated degradation of the mismatch generated by PE and restoring the original sequence. Consequently, compromising MMR protein function, inhibiting EXO1 activity, or blocking the MMR pathway can enhance prime editing efficiency [8,9].

The primitive PE system, referred to as PE1, was developed through the fusion of the Moloney murine leukemia virus (M-MLV) RT with the C-terminus of Cas9 (H840A) nickase. The pegRNA utilized in this system was an extension of the sgRNA, containing a PBS sequence and an RT template. PE1 demonstrated a maximum editing efficiency of 0.7–17% at various loci [8]. Building upon PE1, Liu et al. proposed enhancing prime editing efficacy by modifying the RT component (PE2) (Figure 1A). Compared to PE1, PE2 increased the efficiency of introducing point mutations by 1.6- to 5.1-fold on average, exhibited improved editing efficiency in indels, and was compatible with short PBS sequences, as evidenced by testing various RT mutations.Despite the enhanced efficacy of PE2, there remains a risk of edited insertions being excised due to DNA mismatch repair mechanisms targeting the edited strand. To mitigate this issue during DNA heteroduplex resolution, an additional single guide RNA (sgRNA) was introduced. This sgRNA was designed to complement the edited sequence introduced by the pegRNA while not targeting the original allele. It directed the Cas9 nickase component of the fusion protein to nick the unedited strand at a nearby location opposite to the original nick site. This improved prime editor was designated as PE3, exhibiting approximately three times the editing efficiency of PE2 but causing a higher rate of indels [8] (Figure 1C).

Motivated by the observation that pretreating cells with siRNAs targeting MMR can enhance PE editing efficiency, a novel strategy was proposed for the concurrent delivery of prime editors and MMR repressor units. This transient expression of the MMR repressor protein MLH1d in the PE2 and PE3 systems, now referred to as PE4 and PE5, respectively, significantly improved editing efficiency [9,10].

Chen et al. introduced PE4 (PE2 + MLH1 dominant negative, dn) (Figure 1B) and PE5 (PE3 + MLH1dn) (Figure 1D), increasing gene editing efficiency by transiently expressing an engineered MMR inhibitory protein. This resulted in improved efficiencies for substitutions, small insertions, and small deletions [10]. While optimizing the system is crucial, blocking MMR with MLH1dn at any stage presents significant risks, especially for translational applications, as discussed later. Notably, another group employed a similar approach to enhance PE efficiency through the absence of the MMR system [10].

Given their central role in diverse DNA transactions, MMR proteins can have severe consequences on various biological systems when inactivated. We will briefly summarize their key features in the following paragraphs.

The efficiency of various DNA repair processes, particularly the MMR system, should be a key consideration when designing PE experiments. A malfunction in this system can have wide-ranging consequences, both for individual cells and for the entire body. Notably, one major consequence is an increased level of point mutations across the genome due to unrepaired errors in DNA synthesis [11,12].

*MLH1* is a key DNA mismatch repair (MMR) gene that plays a critical role in safeguarding genomic stability and preventing cancer development. MMR is a process that meticulously corrects errors arising during DNA replication (Figure 2). Uncorrected errors can lead to mutations that fuel cancer development. When mutations occur in MLH1 or other MMR genes, the entire MMR system becomes compromised, resulting in the accumulation of microsatellite instability (MSI) [13,14]. Microsatellites, consisting of repeated sequences of 1–6 base pairs, are abundant in higher organisms’ genomes and show significant genetic variation. Mutations in these regions typically involve the insertion or deletion of a few individual or repeated units, mirroring a pattern of gradual mutation accumulation. While the MMR system typically corrects most microsatellite mutations, cells lacking this system experience a drastic increase in mutation rate [15,16].

## 2. Potential Molecular and Cellular Risks Associated with MLH1 Disruption

### 2.1. Mitochondria

The critical role of mitochondria, ubiquitous organelles within eukaryotic cells, lies in their ability to generate cellular energy (adenosine triphosphate, ATP) via oxidative phosphorylation. Disruptions in mitochondrial function are increasingly recognized as contributing factors in various pathologies, including cancer [17,18]. A recent interesting finding suggests a previously unknown function for MLH1 within the mitochondria [19]. MLH1 exhibits robust mitochondrial localization, and its deficiency triggers synthetic lethality when combined with the inhibition of specific mitochondrial genes, such as POLG and PINK1. Moreover, MLH1 loss is associated with a decline in the oxygen consumption rate and reduced spare respiratory capacity [20,21].

*MLH1*, along with its partner proteins, plays a crucial role in maintaining mitochondrial function. Therefore, targeting these functions suggests a promising alternative therapeutic approach for MLH1-deficient diseases [22]. Interestingly, *MLH1* frequently associates with POLG, a mitochondrial replication enzyme responsible for both safeguarding mitochondrial homeostasis and initiating apoptosis. Notably, disruptions in this interaction appear to be associated with the development of diabetic retinopathy, further highlighting the critical role of MLH1 in ensuring mitochondrial integrity [19,22,23].

### 2.2. Autophagy

Autophagy (“self-eating”) is a cellular recycling process that breaks down unnecessary or damaged components, like proteins and organelles, for degradation in lysosomes or vacuoles. Mutations in autophagy genes have been linked to various human disease [24].

*MLH1*, a key regulator of autophagy signaling, plays a crucial role in cellular survival. Studies have shown that MLH1 promotes autophagy, a self-preservation mechanism, while hindering apoptosis, a form of programmed cell death [25]. This allows cells to resist chemotherapy by activating the DNA mismatch repair (MMR) pathway through the mTOR/S6K1 signaling cascade [25]. Additionally, MLH1 is essential for autophagy activation in response to the chemotherapy drugs, acting alongside the p53 protein, Importantly, inhibiting MLH1 or the ATM-AMPK pathway diminishes autophagy, highlighting their significance in this process [26,27].

Despite long-standing beliefs that autophagy and DNA repair are distinct processes, it is increasingly apparent that they are closely related [27,28]. In other words, the cell ability to function efficiently and competently depends on mechanisms that regulate autophagy and DNA repair. The malfunction of either can result in dysregulation of the other, and vice versa, thus having adverse consequences for cells [29]. The MMR system triggers autophagy in tumor cells, which prevents chemotherapy-induced apoptosis if deficient. In addition, patients with DNA repair defects are thought to have problems with autophagy, leading to premature aging, developmental problems, and neurodegenerative diseases [27].

### 2.3. Variability in Cellular Responses to MMR Defects across Different Tissues

Although all human cells carry out common processes essential for survival, within their tissue environment, they also manifest unique functions contributing to their phenotype. These processes, both common and tissue-specific, are ultimately regulated by gene regulatory networks, which modulate gene expression and its extent [30]. Indeed, a single gene may exert different effects on cellular phenotype across different tissues.

According to Chen et al., Mlh1d-treated cells did not exhibit microsatellite instability (MSI) in the genome. However, clear evidence exists that tissue-specific haploinsufficiency of MLH1 causes MMR-associated cancer [31]. This explains why MMR-related cancers primarily affect the gastrointestinal tract. Therefore, tissue-specific investigations are crucial to analyze which organs or cells are at risk following MMR failure [32].

Additionally, MMR proteins have been shown to inhibit homologous recombination (recombination between similar or identical DNA sequences) [33]. In cells with MMR defects, recombination rates become dramatically higher, leading to gene conversions at the recombined sites, which can promote genome instability and tumorigenesis [34].

### 2.4. Folate

Folate, a key player in the methylation cycle which helps maintain healthy cells, appears to have a complex role in colorectal cancer. Low dietary folate intake has been linked to an increased risk of abnormal cell growth (neoplastic transformation) in the colon [35,36]. This may be due to its role as a methyl donor. One way folate deficiency can impact the colon is through methylation of the MLH1 gene promoter [37]. MLH1 is a critical component of the DNA repair system, and its inactivation through methylation has been implicated in the development of colon and lung cancers [38,39].

It is worth noting that the MMR system has been directly linked to folate-induced apoptosis [40]. Moreover, folate pathway genes might contribute to fertility complications in idiopathic (unexplained) infertility cases [41,42]. Notably, studies show that folate is the most common vitamin deficiency in general population diets [43]. Therefore, any alteration in the MMR system could potentially impact a large portion of the population, making this relevant for translational applications.

### 2.5. microRNA

MicroRNAs (miRNAs) are small, non-coding RNA molecules that regulate gene expression by binding to messenger RNA (mRNA) molecules and preventing them from being translated into proteins miRNAs are involved in various cellular processes, including metabolism, proliferation, cell cycle control, apoptosis, and autophagy [44]. Their dysregulation is linked to numerous human diseases, including diabetes, cancer, and cardiovascular diseases [45,46]. Moreover, making a connection between microRNA (miRNA) and MLH1 holds significant value. A study revealed an intriguing feedback loop between miRNAs and the MMR system. The MLH1-PMS2 heterodimer (MutLα) positively regulates both the processing of miR-422a and other miRNAs [44,47]. Further studies are needed to elucidate how specific miRNAs interact with the MMR system to answer key questions.

### 2.6. Wnt Signaling Pathway

The Wnt signaling pathway plays a crucial role in regulating cell growth, differentiation, and maintenance [48,49]. Alterations in genes within this pathway, including APC, are frequently observed in different pathological conditions. Notably, MLH1 can indirectly be involved in the Wnt signaling pathway by regulating the expression of other pathway components [50]. The MLH1 gene promoter harbors four specific binding sites for the transcription factor TCF7, a pivotal regulator of the Wnt signaling pathway, which regulates the expression of downstream target genes. When TCF7 binds to these sites, it triggers the transcriptional activation of the MLH1 gene [51]. More recently, a positive correlation between TCF7 and MLH1 expression was reported.

### 2.7. Interaction Networks

Another crucial perspective is considering gene interaction networks. These networks consist of genes linked by edges representing their functional relationships. These edges, called interactions, represent potential physical interactions between gene products, where one can alter or influence another. During genetic interactions, two gene variants collaborate to produce an effect that neither could achieve alone [52,53]. For MLH1, BioGRID, a systems biology database (Biological General Repository for Interaction Datasets), reports 416 interactions and 228 unique interactors [54]. This highlights the complex molecular interactions involving MLH1 within each cell, interactions that further change under different physiological and pathological conditions (Figure 3).

## 3. The Clinical Impact of Defective MLH1

The integrity of the genome is critically safeguarded via the MMR system. Any malfunction within this system can lead to severe consequences, such as the development of cancer (Figure 4). Chen et al. found no signs of MSI in their cell lines. However, carcinogenesis is a multistage and long-term process, and oncogenic mutations (not necessarily MSI) in proto-oncogenes or tumor -suppressor genes can create malignant phenotypes several years after they are introduced [55]. By acting as lesion sensors, MMR proteins cause apoptosis and activate cell cycle checkpoints, contributing to tumorigenesis by allowing unchecked cell division [56].

In this section, we will discuss into the clinical outcomes of MMR deficiency, with a specific emphasis on MLH1.

### 3.1. Colorectal Cancer

Colorectal cancer (CRC) is considered the second leading cause of cancer-related mortalities, resulting from the accumulation of genetic alterations in oncogenes and tumor suppressor genes (TSGs) within the colorectal epithelium [57]. Two primary mechanisms of genomic instability have been identified in the progression of sporadic CRC. The first, called chromosomal instability, arises from a cascade of genetic changes that involve the activation of oncogenes, such as *K-ras*, and the inactivation of TSGs, including *p53*, DCC/Smad4, and APC [58,59,60].The second mechanism, referred to as microsatellite instability (MSI), originates from the inactivation of DNA mismatch repair (MMR) genes, such as MLH1 and/or MSH2, mostly via promoter hypermethylation (MLH1ph). This leads to mutations in genes containing coding microsatellites, such as transforming growth factor receptor II (TGF-RII) and *BAX* genes [38]. MLH1 plays a multifaceted role in CRC development [61]. MSI can accumulate mutations in other parts of the genome, including TSGs. These mutations contribute to the development of cancerous cells. Additionally, MSI allows cells to escape normal growth control mechanisms, enabling metastasis [58,62]. Furthermore, MMR deficiency may impair immune surveillance, allowing tumor cells to evade detection and eradication by the immune system [62].

### 3.2. Gastric Cancer

Gastric cancer (GC), a widespread form of cancer worldwide, is classified into two distinct histological subtypes: intestinal and diffuse, according to Lauren’s classification [63]. Intestinal-type gastric cancer is linked to various genetic alterations [64]. Its precursor, known as intestinal metaplasia, is marked by mutations in the p53 gene, reduced expression of retinoic acid receptor beta, and increased expression of telomerase reverse transcriptase [65,66].

Gastric adenomas, precancerous lesions that can progress to metastatic gastric cancer, also display genetic changes. These alterations include mutations in the *APC* gene, reduced expression of the p27 tumor suppressor protein, and amplification of the *cyclin E* oncohgene [67]. Furthermore, in more advanced GC, *c-ErbB2* is often amplified and overexpressed, TGFBRI expression is decreased, and *p27* expression is completely lost. While mutations in the *hMLH1* gene are uncommon in gastric cancer, microsatellite instability (MSI) is more frequent than in colorectal cancer [68]. Here, MSI is primarily caused by hypermethylation of the *MLH1* promoter; for example, 20% of primary GC patients have been reported to have hypermethylated *MLH1* promoters [68,69].

### 3.3. Glioblastoma

Glioblastoma (GBM) is the most frequent and most malignant primary brain tumor in adults [70]. Growing evidence suggests DNA mismatch repair (MMR) gene expression may also be associated with the tumor response to alkylating agents. Notably, *MLH1*, *MSH2*, and *MSH6* are considered intriguing candidate genes [71]. MMR deficiency, frequently observed in GBM, may contribute to therapy resistance and tumor recurrence. Notably, expression of *MLH1* and *PMS2* is reduced in recurrent GBM tumors compared to primary tumors [72,73].

### 3.4. Endometrial Cancer

Endometrial cancer (EC) is a prevalent gynecological malignancy, ranking fourth among women after breast, colorectal, and lung cancers [74].Two distinct types of endometrial carcinoma have been identified, exhibiting distinct pathological and clinical features. Type I carcinoma, linked to hyperestrogenism, commonly manifests as endometrial hyperplasia and is characterized by the frequent expression of estrogen and progesterone receptors, typically affecting younger women. In contrast, type II carcinoma, unrelated to estrogen, is associated with an atrophic endometrium, the infrequent expression of estrogen and progesterone receptors, and a higher prevalence in older women [75]. The distinct morphological features of these cancers are reflected in their molecular genetic profiles. Type I demonstrates defects in DNA repair mechanisms and mutations in *PTEN*, *K-ras*, and *beta-catenin*, while type II exhibits aneuploidy, p53 mutations, and amplification of the *HER2/neu* gene [76].

The underlying molecular dynamics of MMR-deficient/MSI-high endometrial cancers involve three primary classes: (*MLH1ph*), accounting for approximately 70–75% of cases; somatic mutations in *MLH1*, *MSH2*, *MSH6, PMS2*, and/or *EPCAM* occurring in 15–20% of cases; and germline mutations in these genes contributing 5–10% of patients [77]. A smaller percentage (around 3%) of ECs are associated with Lynch syndrome, which is inherited through mutations in one of the MMR genes [78]. It is worth mentioning that improved survival with an intact MMR system in EC has been reported by Cohn et al. [79].

### 3.5. Ovarian Cancer

Ovarian cancer (OC) is considered a primary contributor to mortality among gynecological cancers and ranks fifth as a cause of cancer-related mortalities in women [80]. Hereditary predisposition is identified as a risk factor for OC, with mutations in breast cancer susceptibility genes playing a role. Mutations in the MMR genes are the next most common cause of OCs [81].

This is further fueled by multiple studies reporting a prevalence of *MLH1ph* ranging from 10% to 50%, suggesting it’s a common feature in OC [82]. Additionally, research indicates a notable preference for MMR deficiency in endometrioid and serous carcinomas within OCs [83,84].

It should be mentioned that reported the safety of neoadjuvant pembrolizumab for patients with MMR-deficient solid tumors resulting in high rates of clinical outcome has drawn significant attention to the detection of MMR deficiency in tumors [85,86].

### 3.6. Fanconi Anemia

Fanconi anemia (FA) is a genetic syndrome that clinically affects several human systems. It results in progressive bone marrow failure and predisposes individuals to malignancies, particularly in the urogenital area and the head and neck [87].

In addition to the FANCJ-MLH1 interaction, several additional nodes link the FA and MMR pathways, suggesting that this crosstalk has functional importance. These interactions include BRCA1-MSH2, FANCD2-MLH1/MSH2, SLX4/FANCP-MSH2, and FAN1-MLH1. However, the full functional significance of these interactions remains unclear [88]. MSH2, a key protein in the MMR pathway, plays a critical role in the FA pathway by promoting the localization of FA core components to chromatin, and recruiting FANCJ to sites of DNA crosslinks [89]. Given the relationship between the FANCJ-MLH1 interaction and MSH2, a central question is whether other FA pathway components also function to balance the activity of MSH2 or other MMR proteins [90,91]. The ability of FA cells to cope with replication stress is linked to proteins in the MMR pathway. Furthermore, the MMR pathway plays a crucial role in the repair of DNA interstrand crosslinks (ICLs), toxic lesions that result in chromosomal instability and disrupt normal transcription, likely due to the direct interaction between FANCJ and MLH1 [92,93].

### 3.7. Fertility

From a fertility perspective, the MMR system plays a crucial role in many organisms by facilitating meiotic crossovers, which are essential for various processes during meiosis [94]. However, this involvement could lead to DNA rearrangements and an increased frequency of exchange between partially homologous sequences in germ cells, potentially causing infertility [94,95]. Spermatocytes deficient in MLH1 exhibit premature chromosome separation and arrest during the first meiotic division. Furthermore, human variants in *MLH1* and *MLH3* genes are associated with aneuploidy, pregnancy loss, and premature reproductive aging [95].

## 4. Conclusions and Perspectives

The genomic DNA of each cell is highly dynamic, undergoing an estimated 55,000 single-strand breaks and 25 double-strand breaks daily [96]. Therefore, even temporary inhibition of a multifunctional protein like MLH1 can have unpredictable consequences for individual cells and the organism. Indeed, the proposition that transient MLH1 inhibition enhances PE efficiency is intriguing but not overly surprising, as prolonged inhibition could result in detrimental effects. Of particular note, the discovery of novel roles for the MMR system is likely, adding a new layer of complexity to understanding how MMR interacts with other cellular components like DNA replication and transcription machinery. These issues all await further intensive studies in the MMR field, which could undoubtedly provide important insights into the system’s many other potential roles in DNA interactions.

There is a lack of literature concerning the consequences of the transient knockdown of MLH1 in various cell tissues, particularly in vivo, prior to any in vivo and clinical applications of PE4 and PE5. Universal molecular investigations, such as whole-genome sequencing and RNA sequencing, are suggested to observe any changes before and after MLH1 knockdown. Additionally, other experiments, such as apoptosis and autophagy evaluations, are recommended. Finally, long-term follow-ups in terms of clinical application is recommended to screen for possible side effects.

In summary, future experiments exploring the application of the new version of PE through MLH1 inhibition or targeting other members of the DNA repair system could lead to unpredictable and severe side effects in various cell types, potentially even including the development of cancers.

## Figures and Tables

**Figure 1 cells-13-00858-f001:**
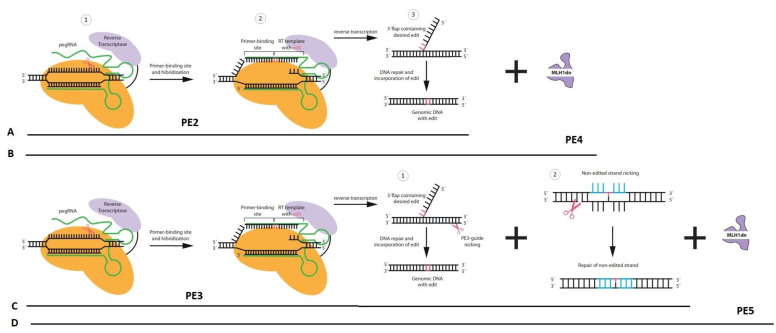
Schematic representation of different types of prime editing (PE2, PE3, PE4, and PE5). (**A**) (1) The PE2 comprises a fusion protein of Cas9 nickase and a reverse transcriptase (RT). This complex is coupled with a prime editing guide RNA (pegRNA). (2) The PE complex binds to the target locus, nicks the PAM strand, and facilitates hybridization between the 3’ end of the PAM strand and the 3’ end of the pegRNA. The primer binding site (PBS) enables the initiation of reverse transcription by the RT. (3) The PE complex dissociates, and cellular repair mechanisms resolve DNA flaps to introduce the edit. (**B**) The PE4 comprises PE2 and a plasmid encoding the dominant negative MMR protein MLH1 (MLH1dn). The PE4 process mirrors PE2 but with higher efficiency due to the inhibition of MLH1 action, reducing the cellular MMR response. (**C**) The PE3 consists of PE2 and an additional sgRNA. (1) This sgRNA is designed to match the edited sequence introduced by the pegRNA but not the original allele. It directs the Cas9 nickase to cut the unedited strand at a nearby site, opposite to the original nick. (2) Nicking the non-edited strand causes the repair system to copy the information from the edited strand to the complementary strand, permanently installing the edit. (**D**) The PE5 consists of PE3 and a plasmid encoding MLH1dn which increases the editing efficiency.

**Figure 2 cells-13-00858-f002:**
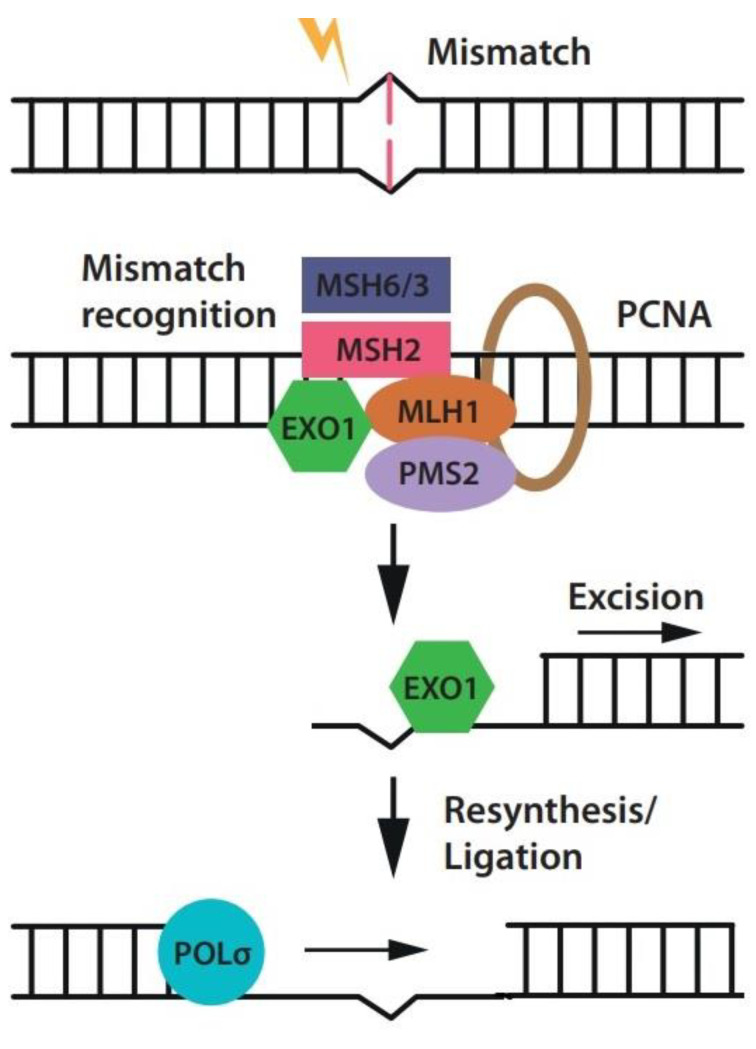
An overview of the MMR process. The DNA damage is recognized by the MutSa (MSH2-MSH6) or MutSβ (MSH2-MSH3) complex following by the MutLa (MLH1-PMS2) complex attachment. Then, exonuclease 1 (EXO1) removes the impaired nucleotides. Finally, DNA polymerase δ (Polδ) incorporates accurate nucleotides.

**Figure 3 cells-13-00858-f003:**
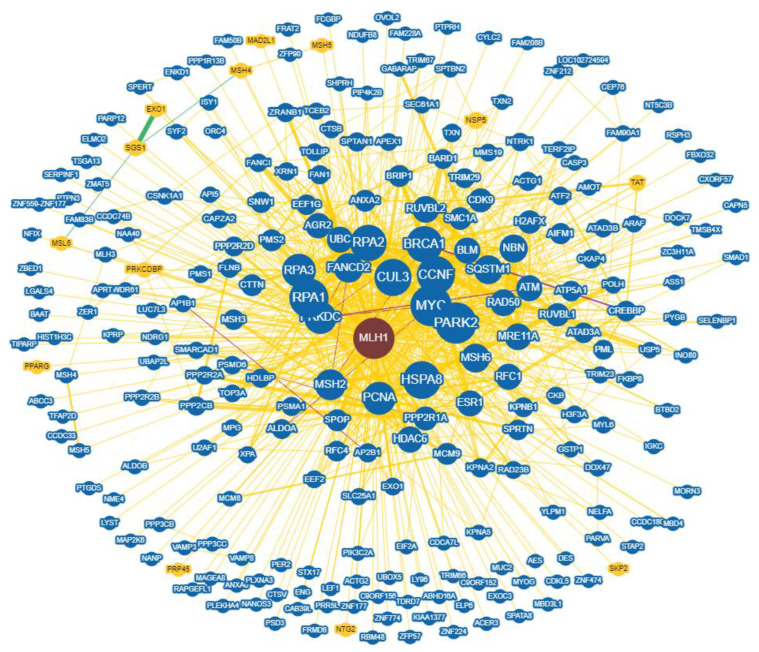
Beyond the basics: MLH1 dynamic interactions beyond mismatch repair through 416 interacting partners.

**Figure 4 cells-13-00858-f004:**
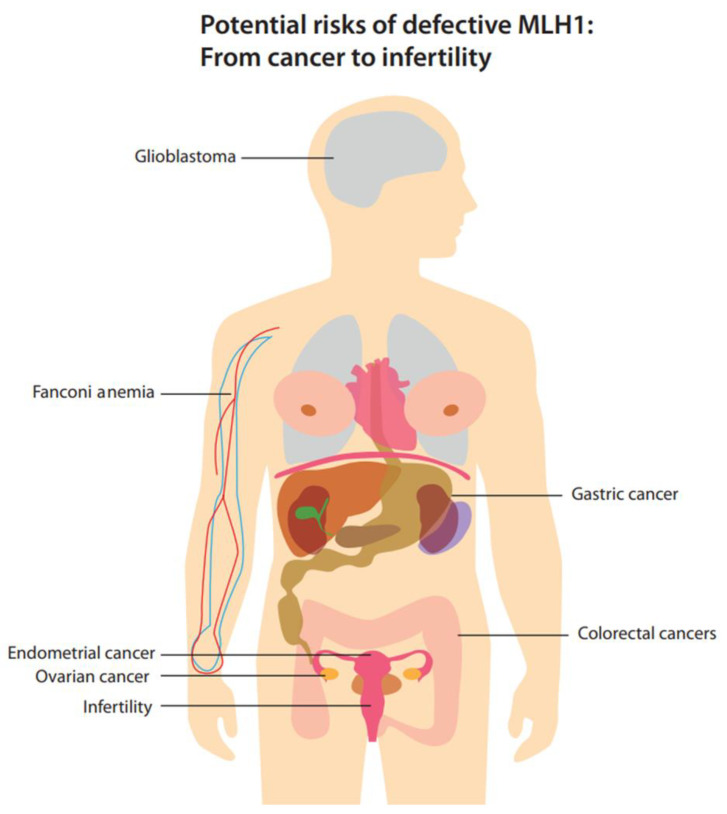
The potential risks of a defective *MLH1* gene, including cancer and infertility.

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
