# Peer review of "Prime Editing and DNA Repair System: Balancing Efficiency with Safety"

_cells, 2024, doi:10.3390/cells13100858_

Round 1
Reviewer 1 Report
Comments and Suggestions for Authors
The review manuscript by Karim et al. discussed the prime editing and DNA repair system with a focus on the safety issue related to the inhibition of mismatch repair factor MLH1. Prime editing is a new CRISPR technology that has significantly expanded the gene editing capability while does not depend on double strand DNA break. Prime editing efficiency has been improved by the manipulation of DNA repair pathways, but it is of importance to realize the potential safety issues as alternation of the natural cellular DNA repair pathway can cause detrimental effect especially for therapeutic applications. While this review manuscript attempts to point this out, there are several points that do not make it qualified for publication in Cells.
1. Gene editing including prime editing in most cases are performed in a transient manner, especially for therapeutic settings, thus the time period of MLH1dn in cell will be short. The addition of MLH1d would only transiently “knockdown“ the mismatch repair system. This is different from the concept of defective genotype, which has been depicted in a large panel in this manuscript;Besides, in line 122, the authors mentioned that in the original article by Chen et al. there is no sign of MSI, which seems to indicate there is no side effect caused by the MLH1dn addition, likely due to its short exposure?
In short, these two points vastly reduces the novelty of this manuscript and might not be of interest to readers, unless there is sign of safety issues caused by short inhibition of DNA repair pathways.
2. The introduction to prime editing is too simple and the history of prime editing development should be depicted in a more details, e.g. what is PE2 and PE3.
3. Line 123 to 125: “Chen et al. found no signs …after they were introduced.” If there is no signs of MSI, there will no malignant phenotypes for sure no matter how many years, right? I do not follow this sentence, please explain.
4. Line 147: What does that mean by “Tissue-specific”? It is not a proper title wording and should be rephrased.
Author Response
I have attached a word file

Reviewer 2 Report
Comments and Suggestions for Authors This is a timely, relevant review, with important considerations for researchers in the prime editing space hoping to bring their research to the clinic. In the first section, it may help to explain the forms of prime editing (PE2, PEmax, PE3) in the text, as well as just in the figure, and to refer to the figure more in the paragraph starting on line 56. On line 81, the reference number does not match the study described. The introductory paragraph for section 2 does no seem to fit with the content of section 2, and may be better located in section 3, with an introduction to section 2 which flows into the content. In section 2, there are multiple areas where relevant molecular detail is lost as to what role MLH1 plays in the specific processes, resulting in rather vague paragraphs. Some examples of this are on line 130, 158, 170, as well as line 215 in section 3. It would be useful to explain the role of MMR or MLH1 in more molecular detail. In section 2.6 on line 179, the CRC abbreviation is used when it is not explained until section 3. In 2.6, line 180, the example given does not seem to match the explanation. It is stated that TCF7 regulates both MLH1 and Wnt signalling pathway, but does not explain how MLH1 affects Wnt signalling. In general, section 3 is well written and has an appropriate level of detail, with interesting points. This review would potentially benefit from an additional section commenting more in depth as to whether transient MLH1dn expression is comparable to to the effects of long term mutations in the MMR, and possibly what approaches the author would recommend for measuring the impact of MLH1dn expression.Author Response
I have attached a word file

Round 2
Reviewer 1 Report
Comments and Suggestions for Authors
The authors have addressed all my questions and the manuscript has now been much improved.